# Construction of an Industrial Knowledge Graph for Unstructured Chinese Text Learning

**Mingxiong Zhao** , **Han Wang, Jin Guo, Di Liu, Cheng Xie \*** , **Qing Liu and Zhibo Cheng**

School of Software, Yunnan University, Yunnan 650500, China
**\*** Correspondence: chengxie@sjtu.edu.cn; Tel.: +1-818-3897-655

**Abstract:** The industrial 4.0 era is the fourth industrial revolution and is characterized by network penetration; therefore, traditional manufacturing and value creation will undergo revolutionary changes. Artificial intelligence will drive the next industrial technology revolution, and knowledge graphs comprise the main foundation of this revolution. The intellectualization of industrial information is an important part of industry 4.0, and we can efficiently integrate multisource heterogeneous industrial data and realize the intellectualization of information through the powerful semantic association of knowledge graphs. Knowledge graphs have been increasingly applied in the fields of deep learning, social network, intelligent control and other artificial intelligence areas. The objective of this present study is to combine traditional NLP (natural language processing) and deep learning methods to automatically extract triples from large unstructured Chinese text and construct an industrial knowledge graph in the automobile field.

**Keywords:** social network; industry 4.0; industrial knowledge graph; deep learning; industrial big data; intellectualization of industrial information

## 1. Introduction

Industry 4.0 is an intelligent era, which promotes industrial transformation through the use of information technology, such that traditional manufacturing and value creation will undergo revolutionary changes. Industry 4.0 is divided into two main parts: one is the intellectualization of industrial control, and the other is the intellectualization of industrial information. There has been much research on the intellectualization of industrial control that is now relatively mature [1–7]. However, the intellectualization of industrial information is still in the research stage and there are some difficulties, mainly because industrial data are heterogeneous and multisource, and most of them are unstructured data. Therefore, determining how to automatically extract useful information from these unstructured data and integrate them is an important part of the intelligence of industrial information. Taking the automobile industry as an example, services oriented toward the users' experience are an important part of value creation and are becoming increasingly more important; however, it is not just the automobile production information but also the valuable information that can be automatically extracted from user evaluations that can help enterprises improve products and serve users. A recent study of the car market found that China has been the world's largest seller of cars for nine consecutive years. In 2017, China's total vehicle sales reached 28.879 million, more than 11 million ahead of the United States and accounting for a third of global sales. The relevant automobile websites and BBS (bulletin board system) generate a large amount of user data, which are mainly unstructured data without a specific format. The main work of this paper is to extract the structured information automatically from these unstructured Chinese texts and build the knowledge graph of the automobile industry based on the extracted structured information. Data sources as well as NLP (natural language processing) or other methods with which to process the data are unique among languages, especially for

those belonging to different language families. Currently, most projects are concerned with knowledge graph systems in the English language. Because Chinese belongs to a different language family, directly translating English knowledge graphs into Chinese is not always feasible; hence, Chinese knowledge graph construction is of great significance. Currently, much progress has been made for knowledge graphs in the English language. However, Chinese knowledge graph construction has more challenges because Chinese is significantly different from English from various linguistic perspectives [8].

In recent years, the knowledge graph, as a new technology to realize large-scale semantic integration and interactive operation, has attracted great attention and research interest from industry and academia. The knowledge graph is a structured knowledge base that is different from the traditional relational database in that a knowledge graph uses a statement composed of two nodes and one edge to represent a fact, which is specifically expressed as a triple (h, r, t) [9], where h represents the head entity, r represents the relationship between the two entities, and t represents the tail entity. A knowledge graph usually consists of a large number of triples. Knowledge graphs have been increasingly applied in the fields of deep learning, computer vision, intelligent control and other artificial intelligence areas. The construction of a knowledge graph is divided into two parts: entity extraction and relation extraction. Knowledge graph has gone through the process from manual construction, such as WordNet and CyC, to automatic acquisition using machine learning and information extraction technology. This paper proposes a novel method that combines entity extraction with relational extraction to realize the automatic extraction of triples that are shaped as "entity-relation-entity" from unstructured Chinese text, and a feasible approach that extracts user evaluation information in the form of "entity-attribute-evaluation" from unstructured Chinese text.

In summary, the contributions of our work are highlighted as follows:

(1) A feasible method is proposed to achieve automatic extraction of triples from unstructured Chinese text by combining entity extraction and relationship extraction.
(2) An approach is proposed to extract structured user evaluation information from unstructured Chinese text.
(3) A knowledge graph of the automobile industry is constructed.

The remainder of the paper is organized as follows: Section 2 reviews the related works. Section 3 describes the proposed method in detail. In Section 4, the complete experiment and the knowledge graph construction is presented. Section 5 concludes the paper.

## 2. Related Work

To construct the knowledge graph of the automobile industry, we need to extract triples, including entity extraction and relation extraction. The related works summarize the state-of-the-art studies about entity extraction, relation extraction, and the introduction of existing knowledge graphs.

### 2.1. Entity Extraction

Entity extraction is also called entity linking or entity annotation. It is a hot topic in knowledge accessing and web-based content processing. Much work has been conducted toward entity linking in recent years, which has resulted in several different solutions. By English entity extraction, Wikify! uses unsupervised keyword extraction techniques to extract entities from text [10]. Then, Wikipedia is applied to find the matching pairs with the extracted entities. Finally, two different disambiguation algorithms are employed to link the correct Wikipedia page with the entity. In a similar way, Tagme and Spotlight extract and link entities to a knowledge base [11–13]. The major difference is that Spotlight uses DBpedia as its knowledge base. For Chinese entity extraction, CMEL builds a synonym dictionary for Chinese entities from Microblog [14]. Then, Wikipedia is applied as the linking knowledge base. An SVM method is used to address disambiguation. Yuan et al. use SWJTU Chinese word segmentation in entity recognition [15]. Pinyin edit distance (PED) and LCS (longest common subsequence) are applied to entity linking. Additionally, Wikipedia is applied as the linking knowledge base. CN-EL uses

a similar process for entity extraction, but the difference is that it uses CN-DBpedia as its knowledge base. It also provides a stable online interface for both research and commercial access. Table 1 summarizes the above methods in detail [16]. It is observed from Table 1 that Wikify! and TAGME are the recommendations for traditional wiki-page linking. Spotlight can be used for LOD linking for English entities and CN-DBpedia can be used for LOD linking for Chinese entities. Recently, entity extraction is transformed into sequence annotation problem, He et al. propose a method about Chinese entity extraction based on bidirectional LSTM networks [17]. Dash et al. use big data mechanics enhance entity extraction [18]. All have achieved good results.

**Table 1.** Summary of entity extraction approaches.

|            | Language | Online API (application programming interface) | Status        | Commercial |
|------------|----------|------------------------------------------------|---------------|------------|
| Wikify!    | English  | Yes                                            | Active        | No         |
| TAGME      | English  | Yes                                            | Active        | No         |
| Spotlight  | English  | Yes                                            | Active        | No         |
| CMEL       | Chinese  | no                                             | update to 2014 | No        |
| Yuan et al.| Chinese  | no                                             | update to 2015 | No        |
| CN-EL      | Chinese  | Yes                                            | Active        | Yes        |

Because of the large number of unrelated entities that would be introduced using the above tools, in this paper, we extract named entities by dictionary matching. We first create a dictionary of the car, and then create a character iterator and identify the name of the car by string matching.

## 2.2. Relation Extraction

Relation extraction is one of the most important tasks in NLP (natural language processing). Many efforts have been invested in relation extraction. Relationship extraction is transformed into relationship classification [19]. One related work was proposed by Rink and Harabagiu [20] and utilizes many features derived from external corpora for a support vector machine (SVM) classifier. Recently, deep neural networks have been shown to learn underlying features automatically and have been used in the literature. The most representative progress was made by Zeng et al., who utilized convolutional neural networks (CNNs) for relation classification [1,21]. While CNNs are not suitable for learning long-distance semantic information, the RNN (recurrent neural network) is often used for text processing [22]. One related work was proposed by Zhang and Wang, which employed bidirectional RNN to learn patterns of relations from raw text data [23]. Although the bidirectional RNN has access to both past and future context information, the range of context is limited due to the vanishing gradient problem [24]. To overcome this problem, long short-term memory (LSTM) units were introduced by Hochreiter and Schmidhuber [25]. Moreover, the GRU (gated recurrent unit) proposed by Cho et al. is a good variant of the LSTM network [26]. It is simpler and more efficient than the LSTM network, so the method of this paper builds on the bidirectional GRU. Most of these methods are supervised relation extraction, which is time-consuming and labor intensive. To address this issue, Mintz et al. align plain text with free-base by distance supervision [27]. However, distance supervision inevitably encounters the wrong labeling problem. To alleviate the wrong labeling problem, Riedel et al. model distant supervision for relation extraction as a multi-instance single-label problem [28], and Hoffmann et al. adopt multi-instance multilabel learning in relation extraction [29,30]. However, all of the feature-based methods strongly depend on the quality of the features generated by NLP tools, which will suffer from the error propagation problem and the difficulty of applying the multi-instance learning strategy of conventional methods in neural network models. Therefore, Zeng et al. combine at-least-one multi-instance learning with a neural network model to extract relations on distant supervision data [31]. However, they assume that only one sentence is active for each entity pair, and it will therefore lose a large amount of rich information contained in those neglected sentences. Hence, Lin et al. propose sentence-level attention over multiple

instances, which can utilize all informative sentences [32]. Since each word in a sentence has a different importance to the semantic expression of the sentence, this paper also uses the word-level attention. In recent years, the research of graph neural network has become a hot topic in the field of deep learning, Zhu et al. use graph neural newtwork extract relation, and achieve good results.

In this paper, we will extract the relation between cars from unstructured Chinese text. For example, given the Chinese text "Volkswagen's two classic b-class cars Magotan and Passat have been occupying a large share of domestic automobile sales", we can extract that the semantic relation between "Magotan" and "Passat" is "Same Level". In this experiment, we define four semantic relations: "Same Level", "Homology", "Subordinate" and "Unknown".

## 2.3. Knowledge Graph

Knowledge graphs can be divided into universal knowledge graphs and industry knowledge graphs. The universal knowledge graph is based on common knowledge and emphasizes the breadth of knowledge. The industry knowledge graph is based on industry-specific data and emphasizes the depth of knowledge. In the universal knowledge graph, Freebase, Wikidata, DBpedia, and YAGO are representative examples. DBpedia is a multilanguage comprehensive knowledge base that was created by researchers from the University of Leipzig and the University of Mannheim in Germany and is at the core of the LOD (linking open data) project [33]. DBpedia extracts structured information from a multilingual Wikipedia and publishes it as linked data on the Internet for online web applications, social networking sites, and other online knowledge bases [34]. YAGO is a comprehensive knowledge base that was built by researchers from the Max Planck institute (MPI) in Germany. YAGO integrates Wikipedia, WordNet, GeoNames and other data sources, and integrates the classification system in Wikipedia with that in WordNet to build a complex hierarchy of categories. Freebase knowledge base was originally created by Metaweb and later acquired by Google [35]. Freebase knowledge base has become an important part of the Google knowledge graph. The data in Freebase is mainly constructed by humans, while the other data are mainly from Wikipedia, IMDB, Flickr and other websites or corpora. Wikidata are a collaborative knowledge base that was designed to support Wikipedia, Wikimedia Commons, and other Wikimedia projects. It is the central repository for structured data in Wikipedia, Wikivoyage, and Wikisource and is free to use [36]. The data in Wikidata are primarily stored as documents and currently contain over 17 million documents. Most universal knowledge graphs are constructed to obtain knowledge from semistructured or structured web pages. In terms of processing semistructured data, the main task is to learn the extraction rules of semistructured data through wrappers. Because semistructured data have a large number of repetitive structures, a small amount of annotation data can allow the machine to learn certain rules and then use the rules to extract the same type of data in the whole site. The construction of an industry knowledge graph is different from the construction of a universal knowledge graph. At present, there is little research on the industry knowledge map and is limited to a few fields. Due to the complex data structure, most of it is unstructured data, which makes the construction of an industry knowledge graph more challenging.

An industry knowledge graph can also be called a vertical knowledge graph. The description target of this kind of knowledge graph is the specific industry domain, which usually relies on the data of a specific industry to build, so its description scope is very limited. In the automotive industry, there is no corresponding knowledge graph. In this article, we will crawl the unstructured data related to the automotive field from the vehicle websites and BBS, and extract the structured knowledge from the unstructured data by employing the method of the bidirectional GRU (gate recurrent unit) combined with an attention mechanism. We construct the knowledge graph of the automotive industry based on the structured knowledge. The construction of the knowledge graph is divided into two main parts: entity extraction and relationship extraction. Entity extraction is also known as named entity recognition (NER) [37] and refers to automatic recognition of named entities from the data set. In this experiment, we automatically extract specific automobile names from unstructured texts, such as "Chevrolet" and "Ford". After entity extraction of the text corpus, we obtain a series of discrete named

entities. To obtain semantic information, we also need to extract the relationship between entities from the relevant corpus and form a network knowledge structure by connecting the entities through the relationship.

Figure 1 shows the pipeline of the method. The input of the method is unstructured Chinese text, where a large number of triples are obtained after processing, and we link the same entities together to form a knowledge graph.

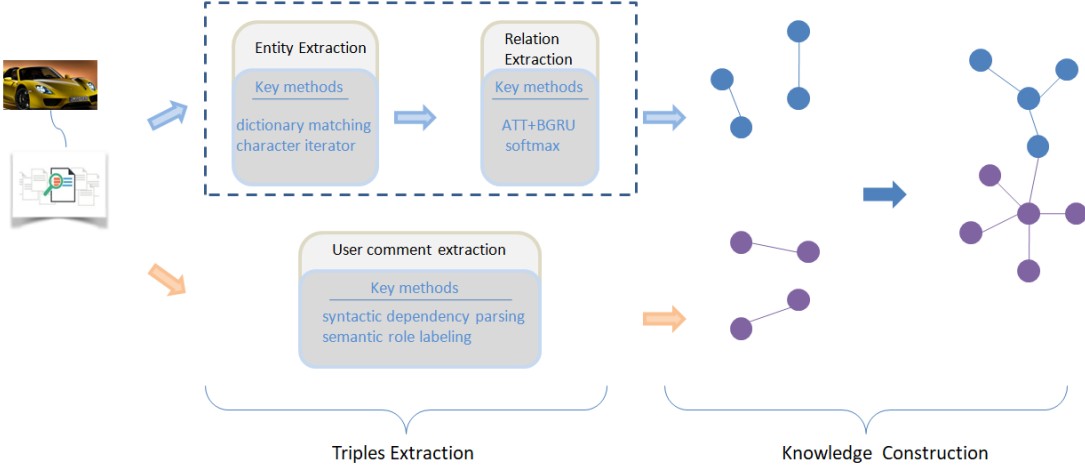

**Figure 1.** The pipeline of the method.

*2.4. Automated Knowledge Base Management*

A fundamental challenge in the intersection of Artificial Intelligence and Databases consists of developing methods to automatically manage Knowledge Bases which can serve as a knowledge source for computer systems trying to replicate the decision-making ability of human experts.

Although the challenge for dealing with knowledge is an old problem, it is perhaps more relevant today than ever before. The reason is that the joint history of Artificial Intelligence and Databases shows that knowledge is critical for the good performance of intelligent systems. In many cases, better knowledge can be more important for solving a task than better algorithms [38].

It is widely accepted that the complete life cycle for building systems of this kind can be represented as a three-stage process: creation, exploitation and maintenance [39]. These stages in turn are divided into other disciplines. In Table 2, we can see a summary of the major disciplines in which the complete cycle of knowledge (a.k.a. Knowledge Management) is divided [40].

**Table 2.** Summary of concepts in the Knowledge management field.

| Knowledge Creation | Knowledge Exploitation | Knowledge Maintenance |
| --- | --- | --- |
| Knowledge acquisition | Knowledge reasoning | Knowledge meta-modeling |
| Knowledge representation | Knowledge retrieval | Knowledge integration |
| Knowledge storage and manipulation | Knowledge sharing | Knowledge validation |

## 3. Methods

*3.1. Semantic Relation Extraction*

The extraction of an entity relationship can be transformed into relation classification. An example is shown in Figure 2, the pipeline of semantic relation extraction mainly includes three steps.

Step one: We convert each word of the input sentence into a vector by an embedding matrix $\mathbf{V} \in \mathbb{R}^{d^w \times |V|}$, where V is a fixed-sized vocabulary and $d^w$ is a hyperparameter to be chosen by the

user. The purpose of providing two entities in input is to calculate the relative distance between each word and two entities, we connect the word vector and position vector to obtain the distributed representation of each word, which is the input of the model.

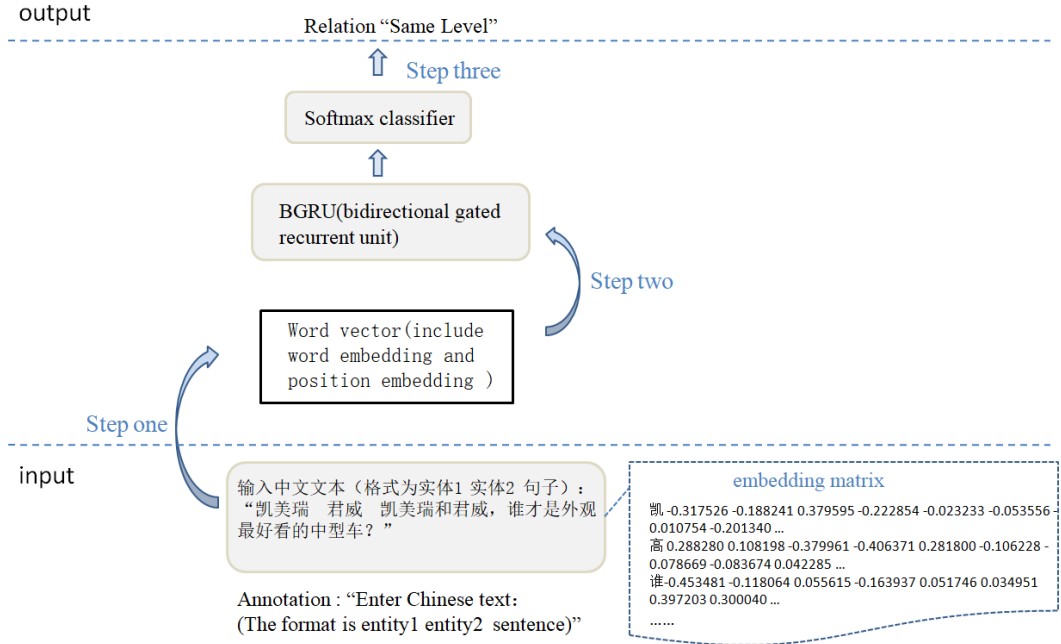

**Figure 2.** The pipeline of semantic relation extraction.

Step two: The model BGRU is able to exploit information both from the past and the future, and finally outputs the distributed representation of the whole sentence.

Step three: After going through the classifier, we can get the probability of each category and select the relationship of maximum probability as the final result.

When we use the model, we simply enter Chinese text (the format is "entity1 entity2 sentences"), and the model outputs relation. Take Figure 2 as an example, we enter the Chinese text "Camry Regal, which is the best-looking midsize car, Camry or Regal?", model output relation "Same Level".

### 3.1.1. Sentence Encoder

In this section, we transform the sentence x into its distributed representation X by the BGRU+Attention model. As shown in Figure 3, the model contains the following components:

1. Input layer,
2. Embedding layer,
3. BGRU layer,
4. Attention layer,
5. Output layer.

The inputs of the BGRU are raw words of the sentence x. We transform words into low-dimensional vectors by a word embedding matrix. In addition, we also use position embeddings for all words in the sentence to specify the position of each entity pair.

**Word Embedding.** Input a sentence x consisting of $n$ words $x = (w_1, w_2, \cdots, w_n)$. This part aims to transform every word into distributed representations that capture syntactic and semantic meanings of the words by an embedding matrix $\mathbf{V} \in \mathbb{R}^{d^w \times |V|}$, where V is a fixed-sized vocabulary and $d^w$ is a hyperparameter to be chosen by the user. As shown in Figure 4, we give a partial word embedding matrix, whose first column is a word, and the latter part is a 100-dimensional vector.

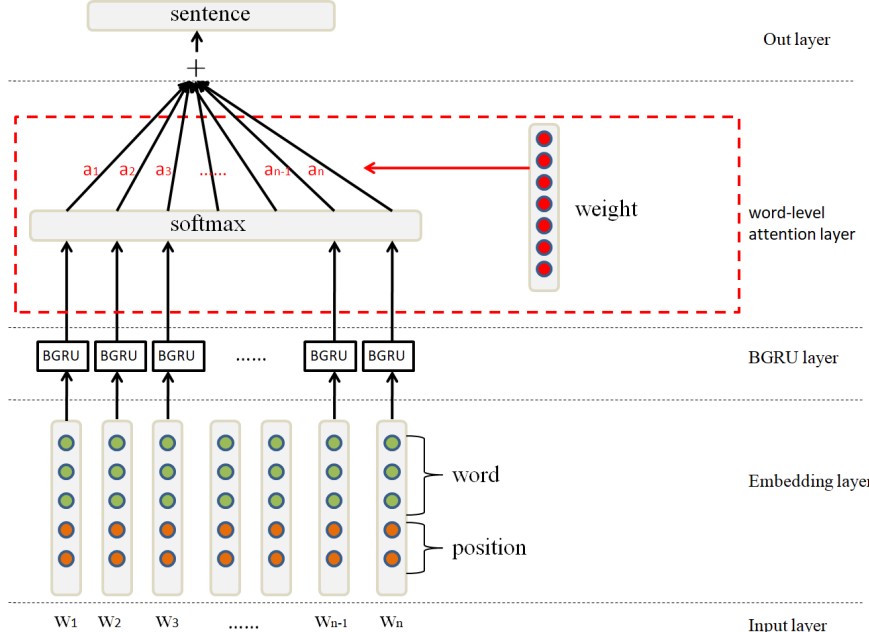

**Figure 3.** The architecture of BGRU (bidirectional gated recurrent unit)+Attention used for the sentence encoder.

Annotation:
| | | |
|---|---|---|
| kay | 凯 | -0.317526 -0.188241 0.379595 -0.222854 -0.023233 -0.053556 0.010754 -0.201340 ... |
| High | 高 | 0.288280 0.108198 -0.379961 -0.406371 0.281800 -0.106228 -0.078669 -0.083674 0.042285 ... |
| Who | 谁 | -0.453481 -0.118064 0.055615 -0.163937 0.051746 0.034951 0.397203 0.300040 ... |
| Steam | 汽 | -0.118446 0.062470 0.194381 0.720970 0.308287 0.142574 -0.136900 -0.161334 -0.167365 ... |
| Ride | 骑 | -0.342828 0.018589 0.178869 0.183846 -0.162444 -0.102450 0.124333 -0.056581 0.192219 0.032627... |
| Man | 郎 | -0.230347 0.004020 0.005433 0.301461 0.043111 -0.317013 0.066877 0.193077 0.387688 0.084718 ... |
| Hugh | 休 | -0.154594 0.133991 0.385062 -0.008949 -0.140869 -0.150756 -0.511297 -0.331444 0.041822 0.432020... |
| Print | 刊 | -0.079298 -0.379804 -0.446937 -0.006616 -0.288559 -0.121265 0.303156 -0.150285 0.134322 - |
| Centre | 中 | 0.041946 -0.008482 -0.159824 -0.049679 -0.076711 0.167492 0.130599 -0.032046 0.064531 -0.079213 |
| Type | 型 | 0.050682 -0.398424 0.092322 0.475399 0.222126 0.147334 -0.019527 -0.068095 -0.118040 ... |
| Most | 最 | -0.212371 -0.048879 -0.345541 -0.315009 -0.200895 0.250481 0.104016 -0.129055 -0.239268 |
| ...... | | |

**Figure 4.** Word embedding matrix.

**Position Embedding.** Contextual information at any location affects the extraction of entity relationships, and the words close to the target entities are usually informative to determine the relation between entities. Therefore, by defining the combination of the relative distances from the current word to the head or tail entities, the GRU can keep track of how close each word is to the head or tail entities.

Finally, we concatenate the word embedding and position embedding of all words to be a vector sequence S = $(w_1, w_2, \cdots, w_n)$, where $w_i \in \mathbb{R}^d$ $(d = d^w + d^p)$.

The GRU (gate recurrent unit) is a kind of recurrent neural network (RNN) that has also been proposed to solve problems such as the gradient vanishing in long-term memory [26]. Compared with LSTM, there are only two "gates" inside the GRU, and it has fewer parameters than LSTM but can also achieve the same function as the LSTM [41]. Considering the computing power and time cost of the hardware, we will often choose a more practical GRU. The architecture of the GRU block is shown in Figure 5.

Typically, the GRU-based recurrent neural networks contain an update gate $z_t$ and reset gate $r_t$. The update gate is used to control the extent to which the status information of the previous moment is brought into the current state. The larger the value of the update gate is, the more the status information of the previous moment $h_{t-1}$ is brought in. The reset gate is used to control the degree of ignoring the status information of the previous moment $h_{t-1}$. The smaller the value of the reset gate is, the more the status information of the previous moment is ignored, just as these following equations demonstrate:

$$
\begin{aligned}
z_t &= \sigma\left(W_z \cdot [h_{t-1}, x_t]\right), \\
r_t &= \sigma\left(W_r \cdot [h_{t-1}, x_t]\right), \\
\tilde{h}_t &= \tanh\left(W \cdot [r_t * h_{t-1}, x_t]\right), \\
h_t &= (1 - z_t) * h_{t-1} + z_t * \tilde{h}_t,
\end{aligned}
\tag{1}
$$

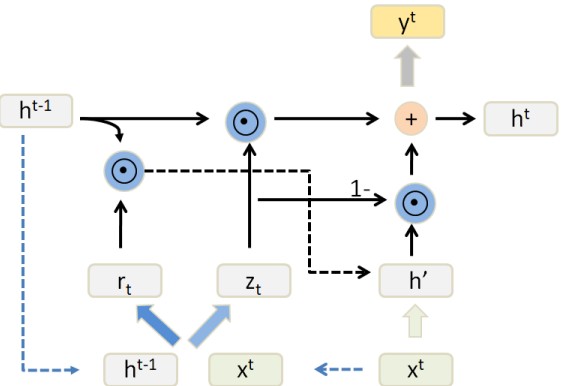

**Figure 5.** The architecture of the GRU(gate recurrent unit) block.

It is beneficial to have access to the future as well as the past context for many sequence modeling tasks. However, standard GRU networks process sequences in temporal order, and they ignore the future context. Bidirectional GRU networks are able to exploit information both from the past and the future by introducing a second layer that reverses the hidden connections flow. As shown in Figure 6, the output is represented as $h_i = \left[ \overrightarrow{l_i} \oplus \overleftarrow{r_i} \right]$.

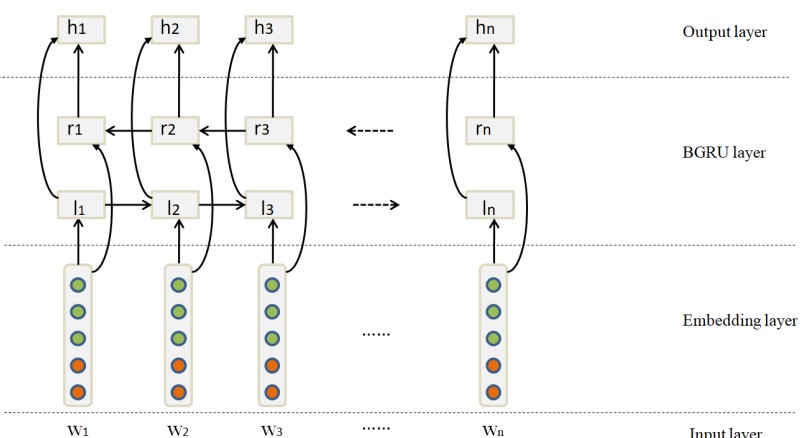

**Figure 6.** The architecture of the bidirectional GRU.

### 3.1.2. Relation Classification

After the embedding layer, the original sentence becomes the corresponding sentence vector. As shown in Figure 7, we use a softmax classifier to predict relation $y$ from sentence set $S$, just as these following equations demonstrate:

$$
\begin{aligned}
p(y|S) &= \mathrm{softmax}(WS + b), \\
y &= \arg\max p(y|S),
\end{aligned}
\tag{2}
$$

where $W$ is a trained parameter vector and $b$ is a bias, and $n$ indicates the number of sentence sets. The loss function is defined as $J$:

$$J(\theta) = -\frac{1}{m} \sum_{i=1}^{m} r_i \log(y_i, \theta), \tag{3}$$

where $r$ is the one-hot representation of the truth relation and $\theta$ represents all parameters of the model.

The attention model was originally applied to image recognition, mimicking the focus of the eye moving on different objects when the person viewed the image [42–44]. Similarly, when recognizing an image or a language, a neural network is focused on a part of the feature each time, and the recognition is more accurate. This motivates determining how to measure the importance of features. The most intuitive method is to use a weight. Therefore, the result of the attention model is to calculate the weight of each feature first and then apply the weight to features.

**Word-level attention.** As shown in Figure 6, the output layer $H$ can be represented as a matrix consisting of vectors $[h_1, h_2, \ldots, h_n]$, where $n$ is the sentence length. The representation $S$ of the sentence is formed by a weighted sum of these output vectors $h_i$:

$$M = \tanh(H),$$
$$\alpha = \text{softmax}\left(w^T M\right), \tag{4}$$
$$S = H\alpha^T,$$

where $H \in \mathbb{R}^{d^w \times n}$, $d^w$ is the dimension of the word vector, and $w$ is a trained parameter vector.

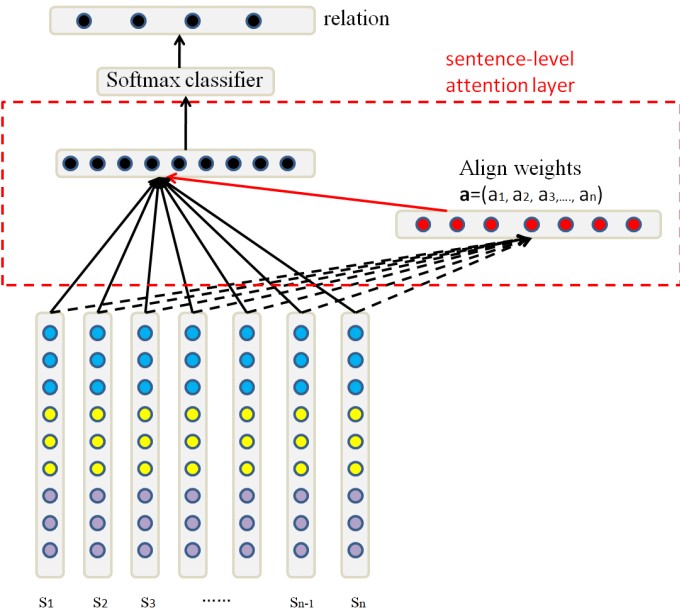

**Figure 7.** Relation classification.

**Sentence-level attention.** As shown in Figure 7, if we regard each sentence equally, the wrong labeling of sentences will introduce a massive amount of noise during training and testing. Therefore, sentence-level attention is important for relation extraction. The set vector $X$ is computed as a weighted sum of these sentence vectors $s_i$:

$$e_i = x_i A r,$$
$$\alpha = \text{softmax}(e_i), \tag{5}$$
$$X = \sum_i \alpha_i S_i.$$

As shown in Figure 8, every line is a sentence (the annotations are in parentheses). Red denotes the sentence weight and blue denotes the word weight. We normalize the word weight by the sentence weight to make sure that only important words in important sentences are emphasized. Figure 8 shows that the model can select the words carrying strong sentiment like "middle-size", "MPV", "same price" and their corresponding sentences. Sentences containing many words like "common", "sales", "from" are disregarded. Note that the model can not only select words carrying strong sentiment; it can also deal with complex across-sentence context.

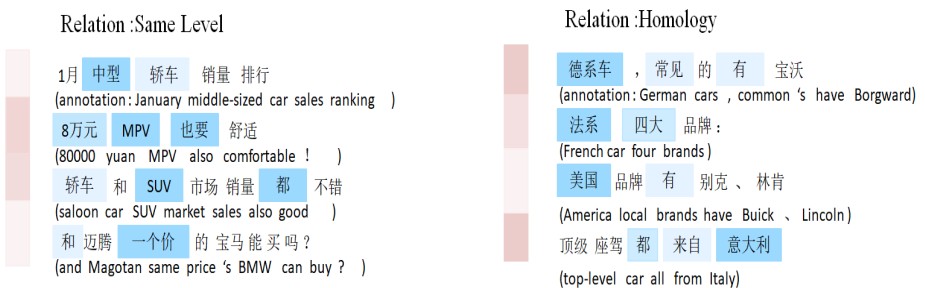

**Figure 8.** Example of visualization of attention.

### 3.2. User Comment Information Extraction

Syntax dependency parsing is one of the key techniques in natural language processing (NLP). Its basic task is to determine the syntactic structure of a sentence or the dependencies between words in a sentence. As shown in Figure 9, an example of syntactic dependency parsing and semantic role labeling is depicted in a Chinese sentence.

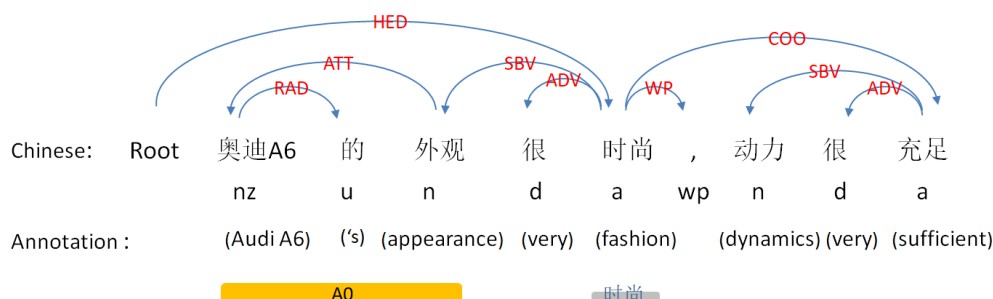

**Figure 9.** An example of syntax dependency parsing and semantic role labeling.

To facilitate subsequent structural understanding and extraction of content, we will organize the results of the above analysis into a dataframe, as shown in Table 3.

The "Word" column is the participle result of this sentence and the "Relation" column represents the relation between this word and the match word. Table 4 shows the corresponding syntactic relations. The "match word" column shows the match terms according to relationships, the "pos" column is the part of speech of each word, the "tuple word" column is a combination of two words, and the "match word n" column is the sequence number of the match word.

Semantic role labeling is a shallow semantic analysis of sentences, which centers on verbs to find the executor and acceptor of actions, as well as the components modified by adjectives. As shown in Figure 10, we find the component "A0" modified by the adjective "fashion" through semantic role labeling, and then find the main component "appearance" and the attribute "Audi A6" through the syntactic analysis of "A0". Finally, we can obtain a triple in the shape of "Audi A6-appearance-fashion".

**Table 3.** An example of dependency parsing.

| | Match Word | Match Word n | pos | Relation | Word | Tuple Word |
|---|---|---|---|---|---|---|
| 0 | appearance | 2 | nz | ATT (attribute) | Audi A6 | Audi A6 - appearance |
| 1 | Audi A6 | 0 | u | RAD (right adjunct) | 's | Audi A6 - 's |
| 2 | fashion | 4 | n | SBV (subject-verb) | appearance | appearance - fashion |
| 3 | fashion | 4 | d | ADV (adverbial) | very | very - fashion |
| 4 | root | root | a | HED (head) | fashion | root - fashion |
| 5 | fashion | 4 | wp | WP (punctuation) | , | fashion - , |
| 6 | sufficient | 8 | n | SBV (subject-verb) | dynamics | dynamics - sufficient |
| 7 | sufficient | 8 | d | ADV (adverbial) | very | very - sufficient |
| 8 | fashion | 4 | a | COO (coordinate) | sufficient | sufficient - fashion |

**Table 4.** The syntactic relations.

| Tag of Relationship Types | Description |
|---|---|
| ATT | attribute |
| RAD | right adjunct |
| SBV | subject-verb |
| ADV | adverbial |
| HED | head |
| COO | coordinate |

### 3.3. Automatic Triples Extraction

We extract named entities by dictionary matching. We first create a dictionary of the car, then create a character iterator, and we identify the name of the car by string matching. Finally, the identified two entities and the corresponding text constitute the input of the relation extraction model. The model outputs the possibility of four relations. We select the relation between two entities with the highest probability, and obtain the triples shaped as "entity-relation-entity". We also obtain the triples of user comments by syntactic dependency parsing and semantic role labeling. Figure 10 shows the flow of triples extraction.

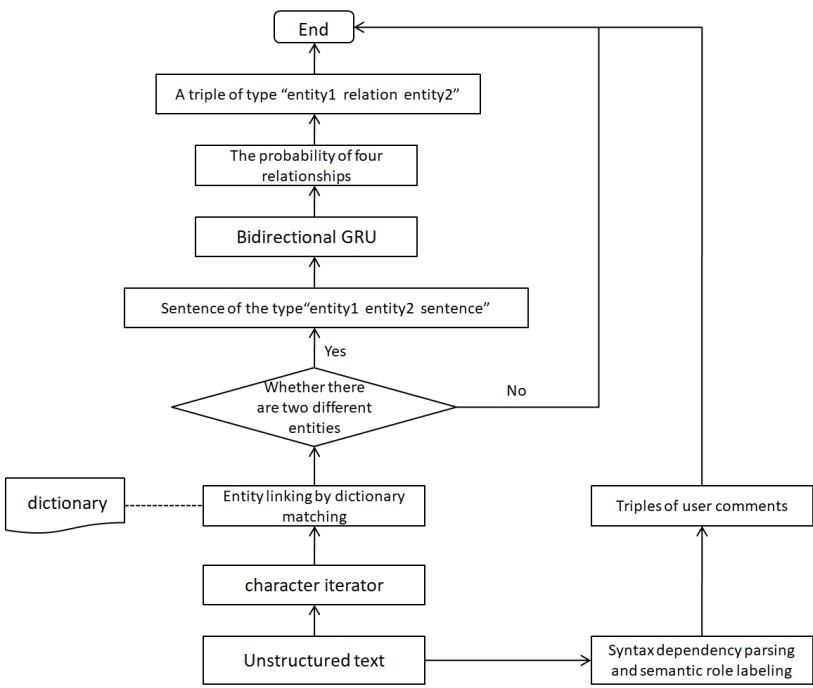

**Figure 10.** Flow chart of triples extraction.

## 4. Experiment

Semantic relation extraction is one of the tasks that is transformed into relation classification and implemented by the supervised learning method in the above section. First, we defined four relationship categories—"Same Level", "Homology", "Subordinate", and "Unknown"—and construct the corresponding data set for each relationship. Then, we train the relationship extraction model and realize the automatic extraction of triples by combining named entity recognition. Finally, we construct the knowledge graph of the automotive domain through the obtained triples.

### 4.1. Dataset

We need to find the corresponding semantic training text for each relationship. For example, the Chinese text "When we talk about French cars, we have to mention PSA group's two twin stars, Citroen and Peugeot" can be expected to be the training text of the semantic relationship of "Homology" between the two entities "Citroen" and "Peugeot". The relationship between two cars from the same country is "Homology". To find the corresponding training text, we first sort out several popular cars from nine countries and then combine the cars from the same country. Finally, we crawl the text in which two cars' names appear at the same time as the training text of the relation "Homologous". The method improves the efficiency of data processing but also introduces considerable noise data. The sentence-level attention mentioned in the previous section reduces the influence of noise data. Figure 11 shows the number of training texts for the relation "Homologous".

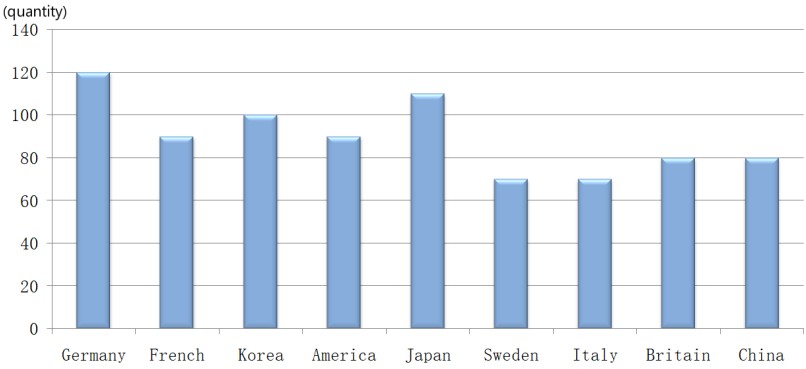

**Figure 11.** The number of training texts for the relation "Homology".

The "subordinate" relationship reflects the information of the superior and the subordinate characteristics. Figure 12 is the subordinate diagram of "Volkswagen", where the relationship between "FAW-Volkswagen" and "Jetta" is "Subordinate". Similarly, we sort out the combinations of other brands and find the corresponding training text for the relation "Subordinate".

Similarly, we sort out 11 levels of partial vehicles, then combine cars of the same level, and finally crawl the corresponding training text. Figure 13 shows the number of training texts for the relation.

Figure 14 shows the data statistics of training data of four kinds of relations, where "unknown" stands for no relation between entities.

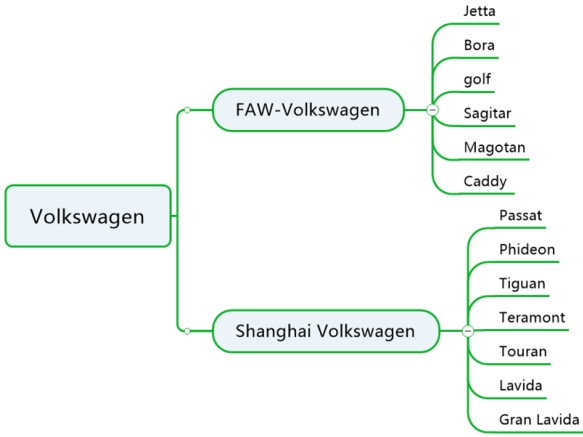

**Figure 12.** The number of training texts for the relation "Same Level".

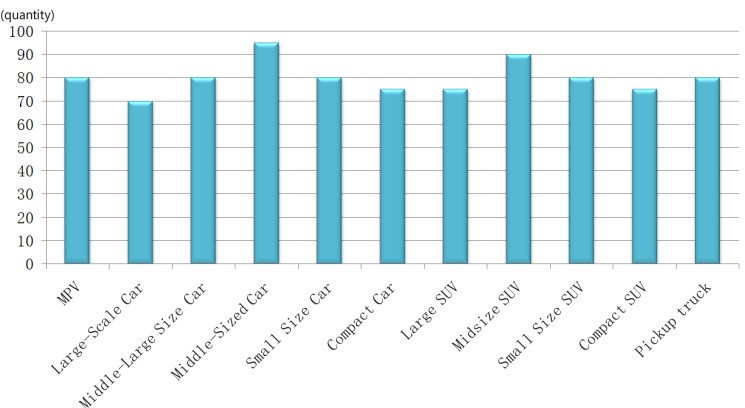

**Figure 13.** Volkswagen affiliate diagram.

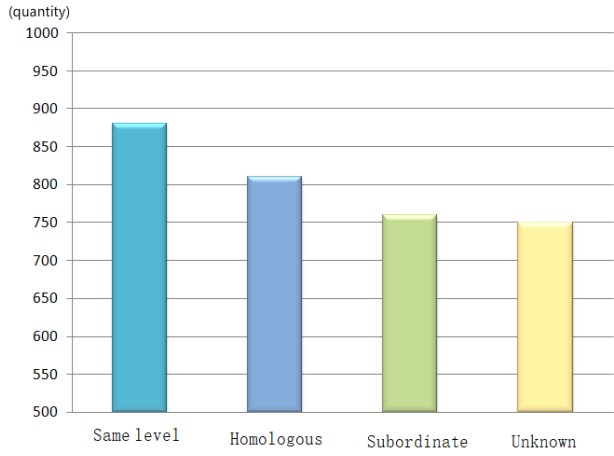

**Figure 14.** Data statistics of training data of four kinds of relations.

*4.2. Model Training*

We train the model with 3200 sentences and test it with 700 sentences. We use a grid search to determine the optimal parameters and select the batch size∈{10,20,...,50}, the neural network layer number∈{1,2,3}, and the number of neurons in each layer∈{200,250,300}. As shown in Figure 15, we form 36 different combinations based on different hyper-parameters, and obtain the average

accuracy of each combination through experiments. Table 5 lists the specific experimental results. We select the hyper-parameter combination with the maximum average accuracy as the optimal parameter set. For other parameters, since they have little effect on the results, so we initialize common values. In Table 6, we show the hyper-parameters used in experiments.

According to whether the classification results are correct, TP, TN, FP, and FN can be determined. TP means that the classification result is a true positive, TN means true negative, FP means false positive, and FN means false negative. We use accuracy and recall rate to evaluate the effect of the model. The specific formula is as follows:

$$P = \frac{TP}{TP + FP},$$
$$R = \frac{TP}{TP + FN}. \tag{6}$$

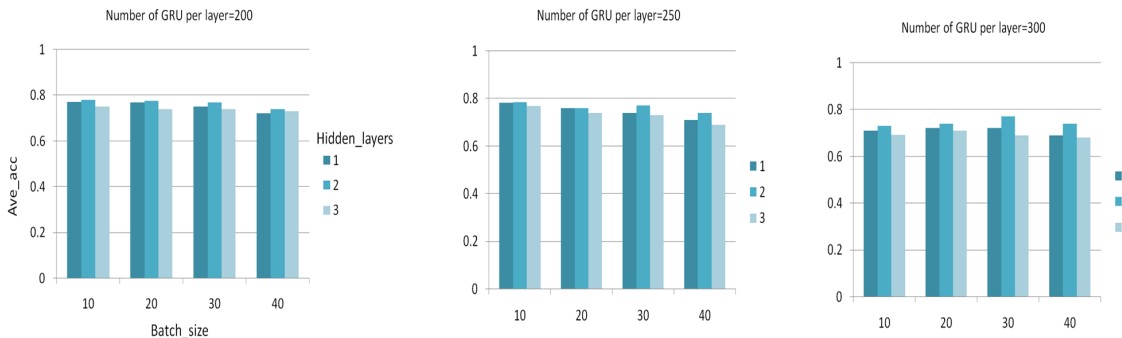

**Figure 15.** Comparison of experimental results of different parameter combinations.

**Table 5.** Average accuracy of different hyper-parameters.

| Number of GRU per Layer | | 200 | 200 | 200 | 250 | 250 | 250 | 300 | 300 | 300 |
|---|---|---|---|---|---|---|---|---|---|---|
| hidden layers | | 1 | 2 | 3 | 1 | 2 | 3 | 1 | 2 | 3 |
| bach size | 10 | 0.77 | 0.78 | 0.75 | 0.783 | **0.79** | 0.769 | 0.71 | 0.73 | 0.691 |
| | 20 | 0.768 | 0.776 | 0.74 | 0.76 | 0.77 | 0.74 | 0.72 | 0.74 | 0.71 |
| | 30 | 0.75 | 0.768 | 0.74 | 0.74 | 0.77 | 0.73 | 0.72 | 0.77 | 0.69 |
| | 40 | 0.72 | 0.74 | 0.73 | 0.71 | 0.74 | 0.69 | 0.69 | 0.74 | 0.68 |

**Table 6.** Hyper-parameter settings.

| | |
|---|---|
| Word dimension | 100 |
| Position dimension | 5 |
| Dropout probability | 0.5 |
| Batch size | 10 |
| BGRU (bidirectional gated recurrent unit) layer number | 2 |
| GRU (gated recurrent unit) size of each layer | 250 |

We randomly divide the data set into training set and test set. We train the model with the training set and evaluate the accuracy of the model with the test set. We divide the data set four times, and carry out experiments for each time. Finally, the average of the results of each experiment is used to represent the performance of the model. As shown in Figure 16, we make a comparative experiment between the two models, the blue curve represents the accuracy/recall rate curve of BGRU, the red curve represents the accuracy/recall rate curve of BLSTM, and the specific results of the four experiments are listed in Table 7.

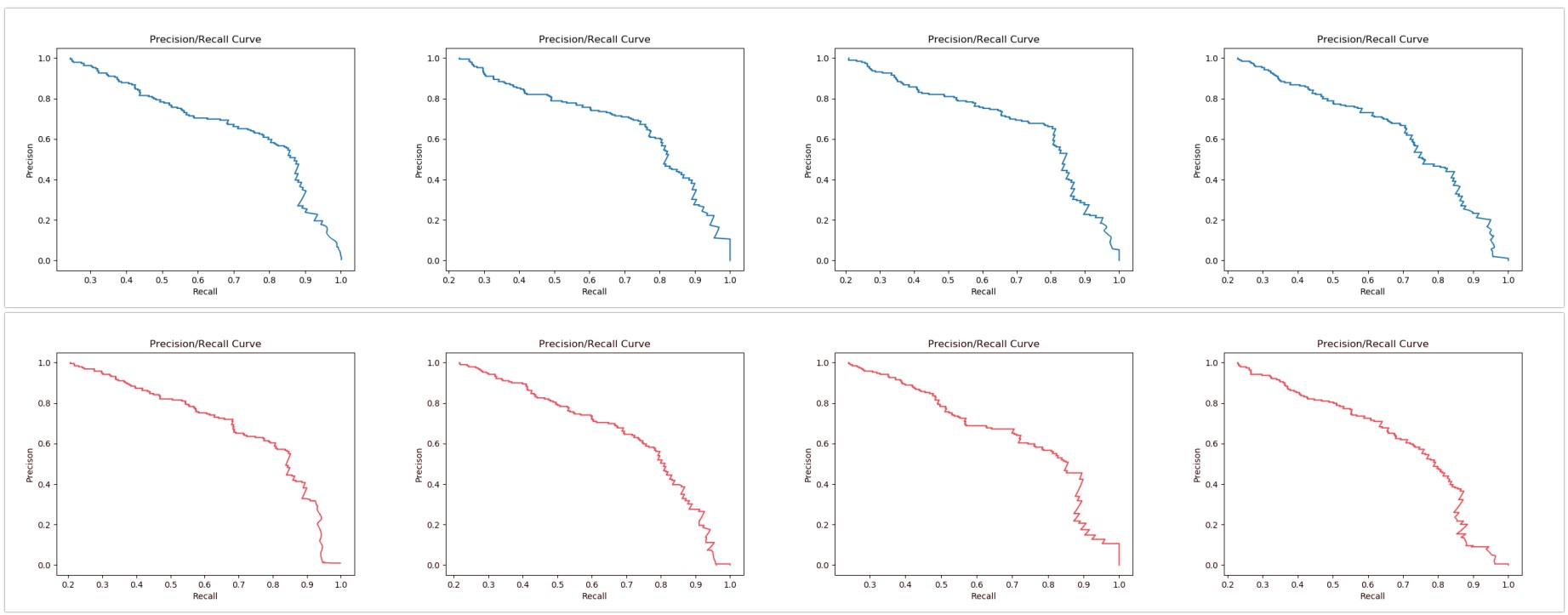

**Figure 16.** Precision/Recall Curve (blue is BGRU (bidirectional gated recurrent unit), and red for BLSTM (bidirectional long short-term memory)).

**Table 7.** The accuracy of four experiments of two models.

|  | 1 | 2 | 3 | 4 | Mean |
|---|---|---|---|---|---|
| BLSTM (bidirectional long short-term memory) | 0.752 | 0.767 | 0.76 | 0.75 | 0.757 |
| BGRU (bidirectional gated recurrent unit) | 0.785 | 0.781 | 0.776 | 0.77 | 0.778 |

We compare the run-time performance of BGRU and BLSTM on a 3.6 GHz Intel Core i7-7700 Think Station P318 with a 32 G DDR4 memory. We calculate the mean values of the four experiments of the two models and comparing the mean values found that BGRU incurs 9.2% smaller run-time compared to BLSTM. Figure 17 shows the run-time of the two models in four experiments. BGRU train faster and perform better than BLSTM on less training data because BGRU has less parameters per "cell", allowing it in theory to generalise better from less examples, at the cost of less flexibility.

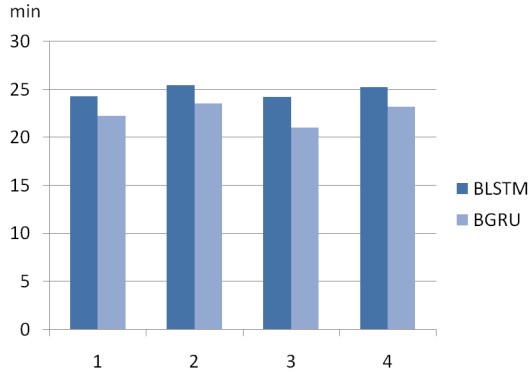

**Figure 17.** Run-time of four experiments of two models.

*4.3. The Result of Triple Extraction*

It is feasible to evaluate the correctness of triples extraction since the test set selected for the work is rather small. Table 8 is the evaluation of car entity extraction in the test data, and evaluation result is given in Table 9. In total, from Table 9, there are 700 texts that could be checked by human efforts. We check these 700 texts and annotate the correct triples as the ground truth. Based on the ground truth, the F1-measure criterion is applied.

**Table 8.** Evaluation of car entity extraction on the test set.

| All Entities | Extracted | Correct | F1-Measure |
|---|---|---|---|
| 1428 | 1400 | 1400 | 0.99 |

**Table 9.** Evaluation of triple extraction on the test set.

| Relation | Ground Truth | Extracted | Correct | Precision | F1-Measure |
|---|---|---|---|---|---|
| Same Level | 230 | 203 | 168 | 0.83 | 0.78 |
| Homology | 150 | 102 | 74 | 0.73 | 0.59 |
| Subordinate | 92 | 63 | 49 | 0.78 | 0.63 |
| Unknown | 228 | 192 | 145 | 0.76 | 0.69 |

**Table 10.** Statistics of triple extraction.

| Texts | Triples |
|---|---|
| 53,200 | 30,500 |

From the result shown in Table 8, almost all the entities in the test set are correctly identified, and the F1-measure achieves 99% calculated by the precision and recall (2*precision*recall/(precision+recall)). It means that most of the car entities in the unstructured Chinese text could efficiently be extracted. It is because the car in both the dictionary and the text share the same naming standard. From the result shown in Table 9, the extraction of triples can achieve more than 73% accuracy, indicating that the model can effectively identify the semantic relationship between entities and extract triples automatically. The triple extraction with the relation of "same level" can achieve a high accuracy rate because the Chinese text about cars often appears keywords that represent vehicle types, such as "SUV,MPV". We believe that cars of the same type satisfy the relation "same level", and these keywords are easy to be given a high weight by the model and easy to be recognized. Similarly, the text also contains some keywords of other relations, and the model can quickly and accurately identify the meaning relations of these texts. However, according to the experimental results, we find that the recall rate of triples extraction with different relations is generally low, which indicates that the extraction efficiency of the model is obviously insufficient for most Chinese texts about cars whose meaning is not clearly expressed, so the model needs more types of texts to train and improve its generalization ability. Finally, we crawl the 50,000+ texts and extract the 30,000+ triples through the model, and Table 10 shows the statistics of the quantity.

Figure 18 shows several examples of triples extraction. An unstructured Chinese text is used as input to the model, and the model automatically outputs two entities and their relation, as well as the triple in the form of "entity–relation–entity".

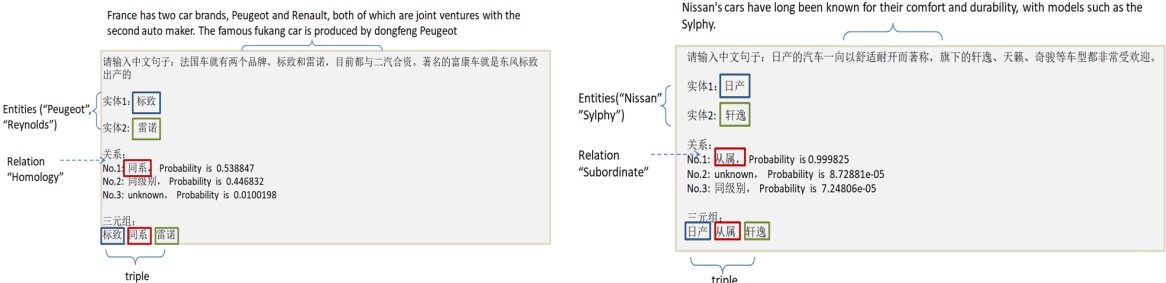

**Figure 18.** Some examples of triple extraction.

### 4.4. Knowledge Graph Construction

As shown in Figure 19, we extract a number of triples from an unstructured text and then link them by connecting entities with the same name. In Figure 20, a knowledge graph composed of partial triples is depicted. The nodes in the knowledge graph represent car entities, and the edges represent the relationship between the two entities. Some Chinese annotations are given in Figure 21.

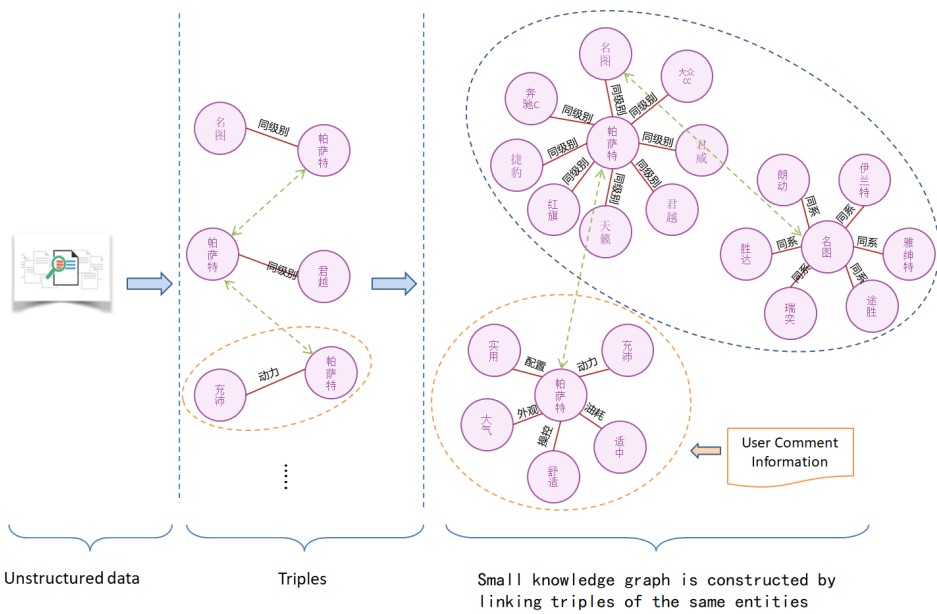

**Figure 19.** An example of the construction of knowledge graph.

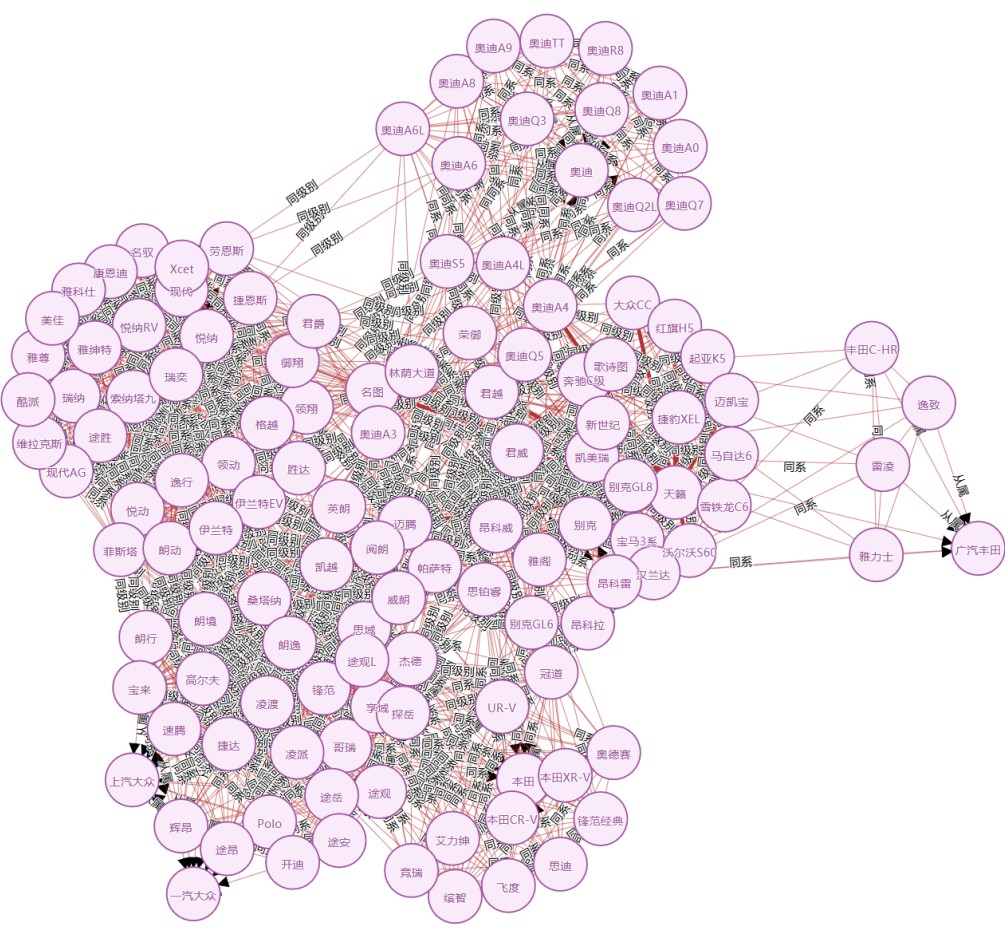

**Figure 20.** knowledge graph of the car.

| node(entity) | | edge(relation) | |
|---|---|---|---|
| Chinese | Annotation | Chinese | Annotation |
| 名图 | MISTRA | 同级别 | Same Level |
| 帕萨特 | Passat | 同系 | Homology |
| 君越 | LaCrosse | 从属 | Subordinate |
| 君威 | Regal | | |
| 大众CC | VWCC | edge(attribute) | |
| 天籁 | Teana | | |
| 红旗 | Red Banner | 动力 | power |
| 捷豹 | Jaguar | 配置 | configuration |
| 奔驰C | Mercedes C | 外观 | appearance |
| 朗动 | Elantra | 油耗 | fuel consumption |
| 胜达 | Santa Fe | ... | ... |
| 瑞奕 | Rui Yi | | |
| 途胜 | Tucson | | |
| 雅绅特 | Hyundai Accent | | |
| 伊兰特 | Elantra | | |
| ... | | | |

**Figure 21.** The partial annotations in the knowledge graph.

*4.5. Discussion*

From the real-world case study, we have learned that the unstructured data in the industrial field contain considerable useful information that can be effectively integrated by the powerful semantic association of the knowledge graph. The method proposed in this paper realizes efficient and accurate extraction of information. On the other hand, three major problems were also learned from the case study:

1. Relation Selection

As mentioned in Table 6, the accuracy and recall rate of triple extraction change with the change of the semantic relation, which indicates that an effective semantic relation setting can improve the efficiency of information extraction. Due to the diversity of industrial fields, entity relations in different fields need to have a special evaluation standard.

2. Entity Extraction

As shown in Table 5, almost all the entities in the test set are correctly identified, which indicates that the dictionary matching method can effectively identify entities in the text. However, this also brings about the same problems. One of them is that the contents of a dictionary need to be complete, and it will take considerable time and money to build a dictionary. In addition, due to the diversity of industrial fields, entity recognition in different fields needs to construct corresponding dictionaries, and this method has poor portability. Entity recognition based on deep learning is more generalized, which is worth studying.

**5. Conclusions**

The industrial 4.0 era is the fourth industrial revolution and is characterized by network penetration. Massive text data will be produced in different industrial fields, but the publication of data are not standardized, and the data quality is not high. The main work of this paper includes:

- A feasible method is proposed to achieve automatic extraction of triples from unstructured Chinese text by combining entity extraction and relationship extraction.
- An approach is proposed to extract structured user evaluation information from unstructured Chinese text.

- A knowledge graph of the automobile industry is constructed.

  In the future, we will explore the following directions:

(1)　We mainly crawl data from BBS and automobile sales websites. We will expand our data in future work, such as unstructured objective data in the automobile manufacturing process or unstructured data in other industrial fields.

(2)　In the process of constructing the industrial knowledge graph, we only aligned the entities with the same name and did not take into account the entities with ambiguity, that is, those with the same name but different meanings. Moreover, we did not merge the entities with different names but which had the same meanings. In the future, we will study the disambiguation and fusion of entities in the process of constructing knowledge graphs.

(3)　We have constructed the knowledge graph of the automobile industry. In the future, we will design a corresponding application according to this knowledge graph. For example, the KBQA (knowledge base question answering) in the automobile field holds prospects.

**Author Contributions:** Conceptualization, C.X.; Data curation, J.G.; Investigation, D.L.; Methodology, M.Z. and H.W.; Resources, Q.L.; Supervision, Z.C.

**Funding:** Science Foundation of Yunnan University: No. 2017YDQN11.; Yunnan Provincial Science Research Project of the Department of Education: No.2018JS008.; Youth Talent Project of the China Association for Science and Technology: No. W8193209.

**Conflicts of Interest:** The authors declare no conflict of interest.

## Abbreviations

The following abbreviations are used in this manuscript:

| | |
|---|---|
| NLP | Natural Language Processing |
| LOD | Linking Open Data |
| GRU | Gate Recurrent Unit |
| CNN | Convolutional Neural Network |
| RNN | Recurrent Neural Network |
| LSTM | Long Short-Term Memory |
| BGRU | Bidirectional Gated Recurrent Unit |
| BLSTM | Bidirectional Long Short-Term Memory |
| ATT | Attribute |
| RAD | Right Adjunct |
| SBV | Subject-Verb |
| ADV | Adverbial |
| HED | Head |
| COO | Coordinate |
| KBQA | Knowledge Base Question Answering |

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
