# Peer review of "Construction of an Industrial Knowledge Graph for Unstructured Chinese Text Learning"

_applsci, doi:10.3390/app9132720_

Round 1

Reviewer 1 Report

Please add more references in 2019.

Reviewer 2 Report

This work is about a novel approach to combine traditional NLP and deep learning methods to automatically extract triples from large unstructured Chinese text. Authors claim that the final goal is to build an industrial knowledge graph in the automobile field.

The work addressed here is interesting. However, there are a number of issues that should be addressed in order to improve the overall quality of the manuscript. These issues are:

- The first sentence of the Abstract and Introduction are the same.
- Lines 33-34, that is not the proper way to reference an existing work in a scholarly article.
- I am not sure on the statement "most knowledge graphs are constructed manually". Do you have any reference to prove that?
- In the state-of-the-art, there is no mention of the techniques for Automated Knowledge Base Management which are in fact the precursors of these new technologies based on knowledge graphs.
- In the bar plots, it is necessary to indicate the units in the Y-axis
- Authors mentiont that they use 2,500 sentences for training. Isn't that too few sentences?
- When connecting entities with the same name, do you use any kind of semantic similarity measure?

Reviewer 3 Report

     The paper presents an interesting enhanced Knowledge graph approach for unstructured Chinese text leaning to improve the accuracy of relationship extraction. 
     In particular, work focuses on combining Natural Language Processing and Deep Learning methods to extract triplets from unstructured Chinese Text.
    This approach based on two main steps method consisting of a first-step the semantic relation discovery that consists of sentence encoder and relation classification using bidirectional gated recurrent unit. The second step consists in achieving automatic triples extraction and Knowledge-based graph construction using natural language processing (NLP). In particular, work focuses on the evaluation of car entity extraction and on the construction of a knowledge graph of the automobile industry to achieve perfect accuracy.

Arguments

Format part: 
   Section 3.1 "Semantic Relation Discovery" is very simple and need to solid illustrations and demonstration to well describe the contribution
    Authors must be improved the section related works to include the revision of the recent literature on this topic.

Technical part: 
   The proposed approach looks sound and interesting. 
   The paper is well organized and presented. 
    However, the experimentation and evaluation parts are weak, Please discuss how to select optimal parameters and need more evaluation results to improve results accuracy, fast execution time, etc.
   The paper proposed review works with several learning techniques from unstructured text. 

    Authors must justify enhanced-BGRU method without any context-aware classification measure in their transformation approach. In addition, authors must show some examples of word embedding matrix and word-level attention matrix.

   The experimental section is the main drawback of the paper; the results need to be discussed in depth taking into account context-aware text classification measure (P & R). We noticed here that all classification measures, but the time complexity not well showed in the proposed algorithm for obtaining optimal parameters.   

Please try to detail how to compare BGRU with BLSTM in terms of time complexity and accuracy.

Thus, the overall suggestion is major revision

Round 2

Reviewer 2 Report

This new version of the manuscript looks much better now. I have still some issues, since the Section about Automated Knowledge Base Management has been included, but no existing works are referenced.

Author Response

Dear Reviewer:

we are very sory for our negligence that not reference existing works. To address the reviewer's concerns, we have added some literature.

[1] Martinez-Gil, Jorge. "Automated knowledge base management: A survey." Computer Science Review 18 (2015): 1-9.

[2] Felfernig, Alexander, and Franz Wotawa. "Intelligent engineering techniques for knowledge bases." AI Commun.26.1 (2013): 1-2.

[3] Rudas, Imre J., Endre Pap, and Janos Fodor. "Information aggregation in intelligent systems: An application oriented approach." Knowledge-Based Systems 38 (2013): 3-13.

Reviewer 3 Report

The revised paper responses to the comments from the first-round review, and it has improved a lot in terms of technical contribution and theoretic analysis.

The reviewer thus recommends to publish the paper as it is.

Author Response

thank you very much!